# Lock-Ins and Agency: Towards an Embedded Approach of Individual Pathways in the Walloon Dairy Sector

**Véronique De Herde [1],* , Kevin Maréchal [2] and Philippe V. Baret [1]**

1   Earth and Life Institute—Agronomy, Université Catholique de Louvain,
    1348 Ottignies-Louvain-la-Neuve, Belgium
2   Gembloux Agro-bio Tech, Université de Liège, 5030 Gembloux, Belgium
*   Correspondence: veronique.deherde@uclouvain.be; Tel.: +32-10-47-37-31

**Abstract:** As the 2009 dairy crisis drew attention to the situation of dairy farmers in Europe, the extent of strategical power left to farmers in dairy cooperatives of increasing size is a frequently raised issue. Four dairy cooperatives collect 97% of the milk in the Walloon Region (in the southern part of Belgium). Two of them integrated agro-food multinationals. We decided to analyze the trajectories of Walloon dairy farmers exploring alternatives to the delivery of milk to these mainstream dairy cooperatives. We focused on the territories situated to the east of the Walloon Region, where dairy farming represents 75% of farming revenues. Alternatives consist either of processing milk on farm or in concluding a contract with a cheese processor collecting milk directly from farmers. Our objective was to understand the issues faced in these alternative trajectories and the reason why these alternatives remained marginal. We designed a qualitative case study based on interviews with farmers and local cheese processors. We mobilized evolutionary approaches on the stability and transitions of systems and approaches of change at the farmer level. It appears that the alternative trajectories remain embedded in a broader dairy context. The lock-ins emerging from this context determine the evolution of the farming model towards intensification and the individual identity and capabilities of farmers. We present a model of interconnected and embedded lock-ins, from the organizational frame of the regime to the individual frame. This model illustrates how the agency articulates with structural dynamics. We propose structural measures in the organization of agricultural education and in terms of support to alternative supply chains that will enhance agency in favor of a change.

**Keywords:** pathways of transition; farmer's identity; cheese processing; alternative pathways; individual trajectories; dairy cooperatives

## 1. Introduction

The year 2009 saw a steep fall of milk price given to dairy farmers, going below 25 cents per liter of milk. As from 2008, following the 2003 Luxembourg Agreement reforming the Common Agricultural Policy, the EU introduced an annual increase in the national milk quotas and a price decline to anticipate the end of the quota system. The link between the European milk prices and the world market prices increased due to these measures. In 2009, the steep decline of the milk world prices [1] induced the so-called "dairy crisis" [2–5].

At the time of the dairy crisis of 2009, in Belgium and neighboring countries, angry dairy farmers shed milk on fields, streets and public institutions and received extensive media coverage [2–5]. The crisis revealed to the public the problematic situation of dairy farmers facing high levels of

indebtedness on their farms [6,7]. The European agricultural policies and the lack of strategical power left to farmers in dairies, especially in dairy cooperatives, has been criticized [8].

The milk sector in the Walloon Region (in the southern part of Belgium—See Figure 1) organizes itself around four dairy cooperatives (further defined as "mainstream dairy cooperatives") collecting up to 97% of the milk produced [9]. Three territories, located in the eastern part of the region (the Région Herbagère Liégeoise, its sub-part called Pays de Herve, and the Haute Ardenne), account for over 40% of the total dairy production while they only represent approximately one-tenth of the entire area of the Walloon Region (Figure 1).

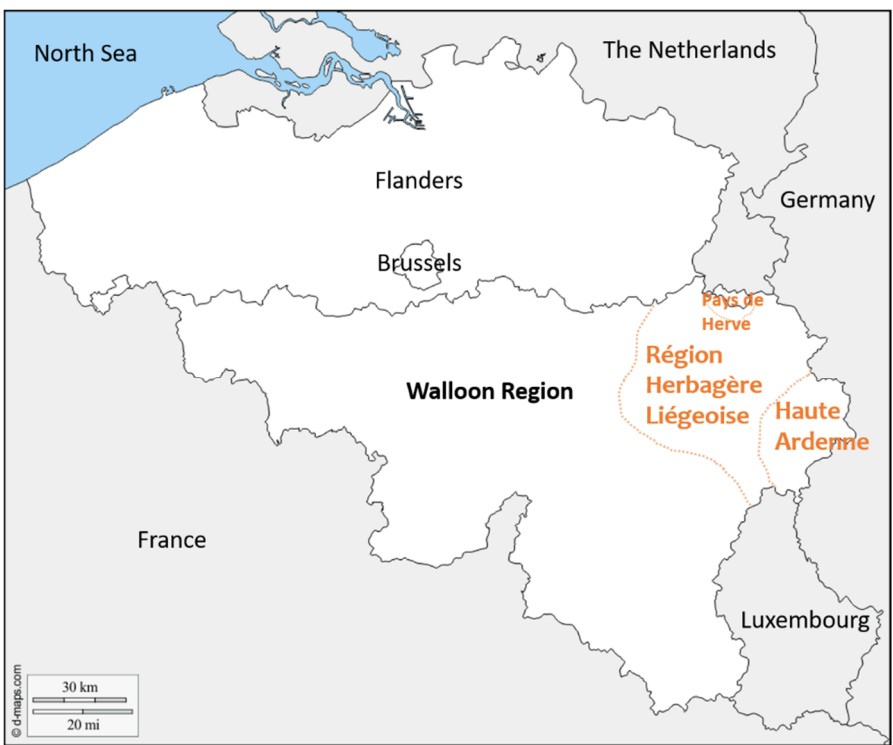

**Figure 1.** Map presenting the location of the Walloon region in Belgium and the situation of the specialized dairy territories.

These territories host one-third of the dairy producers of the Walloon Region, and up to 46% of the specialized dairy farms (farms registered for milk production alone, not in combination with other speculations). Dairy farming represents 75% of the farming revenues generated in these territories. Farms produce milk on grasslands (70–90% of the UAA) and forage crops [10,11]. The dairy farmers of these territories (we further define as "specialized dairy territories") deliver their milk to two mainstream dairy cooperatives. One of the mainstream dairy cooperatives still has a local scale (1900 members over Wallonia) but is mainly active on consumption milk and the production of milk powder. Both products are strongly dependent on the price fluctuations on the world market. The other mainstream dairy cooperative is one of the biggest dairy cooperatives at the European scale, with over 12,000 members. The milk processing strategy of this dairy cooperative is more diversified, but the decentralized position of the Walloon Region within a broader entity does not favor attention to the issues of Walloon dairy farmers.

Could dairy farmers of the specialized dairy territories of the Walloon Region orient the processing of their milk towards productions with a higher added value, and gain more strategical power over the way their milk is being processed? We identified two possible options already present: (1) processing of the milk on farm; (2) making an agreement with a processor who does not collect his/her milk from the mainstream dairy cooperatives but collects his/her milk directly from farmers (further defined as

"local processor"). Milk processing on farm is a marginal practice in the specialized dairy territories: one thousand dairy farms are present [10,11], and only one hundred registered for transformation on farm, mainly for the production of butter [12]. Although these territories have had a past tradition of cheese processing, at the time of the study, only six farmers were carrying out cheese processing on the farm (cow milk) [13]. Concerning the direct delivery of milk to a local processor, we identified only six local cheese processors collecting milk directly from one dozen farmers [13].

The scientific literature stresses the importance of alternative food networks for their transformative potential towards sustainability [14–16]. In this regard, cheese processing alternatives (on farm or by the direct delivery of milk to a local cheese processor) are interesting because most actors do not limit themselves to direct selling and short channels of distribution but also experience distribution through long supply chains, via wholesalers.

We propose a qualitative study in the specialized dairy territories, based on semi-directed interviews with local cheese processors and farmers delivering milk to local cheese processors or producing cheese on farm. In particular, to understand why alternatives to the delivery to mainstream dairy cooperatives do not develop more, we intend to answer the three following questions: (1) Which processes does the farmer face when engaging in cheese processing alternatives? (2) How does the exploration of an alternative channel of milk processing relate to the farming system and the way dairy farmers approach their work? (3) How is their intention towards a change in trajectory supported by the dairy context in which they evolve?

### Changes in Trajectories May Face a Logic of Inertia Inherent to Sociotechnical Systems

Several researchers have pointed to the importance of path dependency and lock-ins to explain the inertia characterizing many sociotechnical systems. The central idea behind these concepts is that dominant routines in production, the use of technologies, knowledge transmission, and institutional and social practices orient future trajectories and hinder other pathways of development at the individual and collective level [17–20].

Following an initial paper from Cowan and Gunby [21], many researchers have applied this set of ideas in empirical studies demonstrating the locked-in nature of agricultural sociotechnical systems. For example, supply chain organization, genetic selection, research and public support policies act in a convergent way and create an unfavorable context for the adoption of fungicide-resistant wheat varieties [22] or the reduction in the use of chemical fertilizers [23]. Production standards [24–27] orient pathways of production and consumption. The organization of supply chains and the imbalance of strategic weight among actors act against the financial support of alternatives [28]. The organization of research and education prevents the development of an integrated approach to production issues [29–31].

Conceptual frames like the Multi-Level Perspective [32,33] consider how socio-technical systems, the "tangible elements needed to fulfil societal functions" [34], co-evolve with a set of rules in a "socio-technical regime" and orient the routines of social groups [34]. In a stabilized regime, lock-ins are, at the same time, the consequence of path-dependent processes and the source of further path-dependency [35,36]. Alternatives to the practices of the dominant socio-technical regime emerge in niches, defined as "constellations with novel, or deviant functioning" [37] or "protective spaces". In niches, innovation develops besides the selective pressure of the socio-technical regime [17]. Typically, if we refer to our specific research, this framework would make us consider the system of the delivery of milk to the mainstream dairy cooperatives as the dominant socio-technical regime, and the alternatives of milk processing on the farm or direct delivery to a local cheese processor as niches.

The frameworks considering the stability and transition of systems [32,33,37] are relevant regarding a retrospective approach of societal changes [18]. In the agricultural sector, these frameworks have been mobilized to assess processes of transition, including recent evolutions towards a more sustainable mindset in agriculture [38,39]. Agriculture and food production are land-based activities, which entail, within a shared mainstream set of practices, strong heterogeneity. Niches may not emerge as coordinated

and separate spheres with transformative ambitions, but emerge from within that heterogeneity [39]. When considering potential transitions in agriculture, and processes of potential transition in the making, the trajectories of individuals are a relevant level of analysis [40].

Change in farmers' individual trajectories is not straightforward. At the farmer level, capital investment, risk evaluation, market configuration, capabilities of the actor act against change or against the ability of the farmer to interpret an event as a trigger for change [35,40]. In addition to lock-ins of a technical and financial nature, knowledge and cultural lock-ins play an essential part [35]. Practical experience and formal education contribute to the emergence of lock-ins, as well as the "the adherence to mutually accepted farming ideals" within the peer group of farmers [41]. The strength of the symbolic value attached to the "good farmer" as a behavioral driver has been stressed in several studies [42–45]. However, emphasizing structural determinism does not help to understand how change happens, and many authors emphasize the importance of considering agency aspects and the impact of the agency on changes [33,36,46–48]. A change in practices implies a continuing process of shifts in meanings that interact with the identity of farmers [36,49,50]. The capability of farmers as a condition and driver for agency and change is a subject increasingly studied in the scientific literature. Capabilities are analyzed in terms of acquired skills [51] but also, in approaches inspired from constructivism [52], in terms of interactions and networks [53,54] and resilience [55,56]. One should also consider the context in which individuals evolve in order to understand how the agency may exert itself [30,36,48,57–60].

Studies focusing on the dairy sector specifically center on the general context and trends of evolution of the dairy sector in Europe and elsewhere [61], or on the way sustainability is integrated at the farm level [62–64] and by the processing actors [65–67]. Concerning individual trajectories and their relation with the dairy context, we identified a few studies focusing on the decision-making processes of dairy farmers in reaction to a certain economic context [68,69], or in reaction to the evolution of public policies [70,71]. In both cases, the focus lies on the strategies of farmers regarding their farm models and the way in which they might make it evolve. By focusing on the individual trajectories as the level of analysis, we intend to understand which contextual factors interact with the individual's ability to consider pathways of change towards a greater diversity of options for the processing of milk in specialized dairy territories of the Walloon Region. Our objective is to understand the issues faced in the alternatives to the delivery of milk to mainstream dairy cooperatives (cheese processing on farm or the direct delivery of milk to a local cheese processor) and the reason why these alternatives remain marginal.

## 2. Materials and Methods

We designed a qualitative study based on semi-directed interviews with the actors active in the above-mentioned alternative trajectories (cheese processing on farm or the direct delivery of milk to a local cheese processor). This approach has been mobilized to study food systems [72], from change at the farm level [22] to social perceptions related to food production [73,74]. The relevance of qualitative approaches for understanding complex systems is now recognized [75,76].

We adopted a "grounded theory" approach [77], taking into account what the data collection revealed beyond any theoretical hypotheses. We fed our interpretation with the help of the described conceptual framework on the stability of systems and change at the individual level.

We placed our focus on the specialized dairy territories of the Walloon Region given the importance of milk in the farming revenues. We identified farmers and local processors in the online data of the regional agency for agricultural promotion [13] and a published guide of Walloon cheese makers [78]. We ensured that our sample was representative of all types of actors present in the studied alternatives: farmers and local cheese processors (Table 1). Fifty percent of the local cheese processors and of the farmers carrying out (or had carried out) cheese processing on farm, eighty percent of the farmers delivering to a local cheese processor, accepted an interview (Table 1). We looked for farmers having

stopped or who refused direct delivery to a local cheese processor. The only one we found refused an interview.

**Table 1.** Qualitative sample distribution for interviews investigating challenges for farmers of the specialized dairy territories of the Walloon region, who process milk on farm or deliver milk directly to a local cheese processor, and for local cheese processors who collect milk directly from farmers.

|  | Local Cheese Processor (Who Collects Milk Directly from Farmer) | Farmer Processing or Having Processed Milk on Farm | Farmer Delivering Milk Directly to a Local Cheese Processor |
|---|---|---|---|
| Identified in the specialized dairy territories of the Walloon Region and contacted for an interview | 6 | 9 | 12 |
| Accepted an interview | 3 | 5 | 10 |

We interviewed five farmers active in cheese processing on the farm (fc-1 and fc-2) or who had stopped cheese processing on the farm (fnc-1, fnc-2, fnc-3). We interviewed three local cheese processors (cp1, cp2, cp3) and ten farmers delivering their milk to local cheese processors (fm-1 to fm-10). Our interviews covered equally the three territories of our geographical study area.

Six of the ten farmers delivering their milk to local cheese processors were of the male gender and worked alone on the farm (fm2, fm3, fm4, fm6, fm8, fm9). The other four farmers delivering their milk to local cheese processors ran their farm as a family business with several members of the family involved (man, wife, sons and daughters). We interviewed the man in two cases (fm1, fm5), and a man and wife in a common interview in two cases (fm7, fm10).

In the case of the farmers processing on farm, farmers ran their farm as a family business too. In one case (fc2), we interviewed a man and wife in a joint interview, in one case the wife (fnc-2), and in the other cases, the man alone (fnc1, fnc3, fc2).

The interviews took place between November 2013 and January 2014. We asked the interviewees to (1) present their activities and their history; (2) identify the factors of success; (3) and the constraints in their trajectories.

The interviews were audio recorded and transcribed. We used the software RQDA to attribute thematic codes [79] to interview parts, and extracted them for analysis. We defined the codes according to our objective and enriched them with elements identified as relevant during data collection. The extracts constituted our material for interpretation.

## 3. Results

### *3.1. Exploring Cheese Processing Alternatives Entails Adaptations Regarding Farm Model and Reveals Lock-Ins Acting against Changes in Pathways for Farmers*

3.1.1. The Requirements Linked with Cheese Processing Influence Farm Model and Practices

A farmer wishing to engage in cheese processing alternatives may choose to transform cheese on the farm. One main obstacle to cheese processing on farm is the absence of familial resources available to add this activity to the running of the farm. When processing on farm is not an option, the farmer relies on the existence of local cheese processors willing to collect his milk.

Both farmers carrying out cheese processing on farm and farmers delivering their milk to local cheese processors adapted their farm practices. The interviewees link the adaptations to requirements in terms of milk properties (taste, protein content, and absence of certain germs). The adaptations also concern the quantity of milk produced during the year. Milk produced on grasslands is more abundant in spring: farmers organize their calving season at this period of the year to support the

lactation peak with the spring grass. However, the demand for cheese is more abundant in winter. Finally, the adaptations concern the distribution of risk between milk suppliers: having numerous small-scale suppliers is less risky than a unique milk supplier.

Table 2 summarizes the practices adopted by the farmers to meet the requirements linked to cheese production. Some practices answer the requirements directly. Other practices answer the requirements indirectly in the way that they offer a better economic efficiency to the farmer.

**Table 2.** Requirements linked to cheese processing influencing the farm model and practices of farmers of the specialized dairy territories of the Walloon Region who process milk on farm or who deliver milk directly to a local cheese processor.

| Requirement | Influenced by | Constraints for Farmer Linked with Requirement | Practice Answering the Requirement * or Providing a Better Economic Efficiency ** |
|---|---|---|---|
| Gustative quality of milk | Feeding | Limitation in the use of concentrates | Extensive milk production ** |
| | | Farmer has to make silages that are less acidic; that is, dryer silages—less nutritional value and higher processing costs (realization of bales necessary) | Extensive milk production. Autonomous realization of clamps (no recourse to sub-contractors to harvest the grass and make the silages, so that the farmer can take the necessary time to ensure a thorough compacting of the dryer silages) ** |
| | Sanitary status of the cow | | Extensive milk production * More rustic cow breeds * |
| Cheese-processing properties of milk | Cow selection— Cow breed | | Selection of another cow breed than the Holstein, or crossings * |
| Sanitary quality of milk | Sanitary status of the cow and feeding | Farmer has to make dryer silages to prevent the development of undesirable microorganisms—Less nutritional value and higher processing costs (realization of bales necessary) | Extensive milk production * Rustic cow breeds * Autonomous realization of clamps (no recourse to sub-contractors to harvest the grass and make the silages, so that the farmer can take the necessary time to ensure a thorough compacting of the dryer silages) ** |
| Distribution of risk among milk producers | Number of milk producers | Farm has to be small scale | Small-scale farm * |
| More milk production in winter | Calving season in autumn | Additional feeding costs linked with the displacement of the lactation peak in winter to answer the needs of the local cheese processor | Extensive milk production—low-input approach regarding feeding ** |

* marks the practices answering the requirement linked to cheese processing; ** marks the practices allowing to answer the requirement with a better economic efficiency.

3.1.2. Lock-Ins Act against Changes in Pathways for Farmers

Interviewed farmers are well aware that their farm model clearly/typically does not follow the broader trend toward large-scale intensive dairy farms, based on the Holstein breed (fc1, fc2, fm2, fm3, fm7, fm8, fm9), also described in the scientific literature [1].

They point out elements that reinforce this trend to large-scale intensive farms (Table 3):

1.　Mainstream dairy cooperatives work with a payment system in function of the quantity delivered by the farmer: they give a bonus payment per liter as from an annual quota of 540,000 L (fm3);
2.　Mainstream dairy cooperatives are more and more reluctant to collect milk from small-scale farms: interviewees mention the fact that small-scale farms turning around 100,000 L a year had been refused collection (fm7, fm3);
3.　The public agricultural advisers encourage farmers to grow in size and invest in equipment. The advisers recommend the use of regional aids dedicated to agricultural investment in the frame of the European rural development program (fc1, fm9);
4.　The loan policies of banks are not favorable to small-scale projects (fm3).

Local cheese processors do not easily find farmers meeting their requirements (gustative, sanitary and cheese-processing quality of milk, and farm size). Local cheese processors look for a farmer whose farm model corresponds to their requirements or who is willing to make the necessary adaptations. This means sometimes driving more kilometers to collect the milk. The interviewees also identify a cultural lock-in acting against the consideration of change: the sense of security linked to mainstream dairy cooperatives. Although this pathway is less satisfactory regarding personal value and remuneration (cp3, fm3, fm7, fm9, fm10), most mainstream dairy cooperatives are "too big to fail": they will benefit from support in case of difficulties. A local cheese processor, conversely, could go bankrupt or decide to reduce the volume of his production (cp2, cp3, fm3). Furthermore, some banks take into account where the farmer delivers its milk before granting a loan, leaving farmers who do not deliver milk to mainstream dairy cooperatives in a situation of uncertainty (fm3).

The interviewees identify the high workload in large-scale intensive farms as a technical lock-in: the attention of farmers is drawn by the sole production of milk, which prevents the consideration of a change in pathway (fm7, cp3). Heavy investments in milking and farm equipment hinder changes in farming or milk processing practices (cp3, fnc2, fnc3) and reinforce the reluctance to leave a mainstream dairy (fm9, fm10).

**Table 3.** Lock-ins identified by the interviewed farmers and local cheese processors of the specialized dairy territories of the Walloon Region, acting to prevent farmers from considering changes in pathways, in terms of farm model and choice of milk processing.

| Lock-Ins Acting against Changes in Pathways of Change by Farmers |
|---|
| Mainstream dairy cooperatives offer bonuses as from a certain quantity of milk and are reluctant to collect milk from small-scale farms |
| Dairy farmers share a common vision about farming practice based on intensification, and the education of farmers contribute to this common vision |
| Public agricultural advisers and banks support farming practices based on intensification, growth and high investment |
| Dairy farmers define themselves as milk producers |
| The high workload on farms and the heavy investments in farm equipment hinder changes in milk processing practices |
| Mainstream dairy cooperatives offer a sense of security |

Interviewees also raise the issue of agriculture schools: they prepare dairy farmers to be milk producers solely (cp3, fm1). Interviewees noted that schools and public advisors advocate for farms

growing in size and following intensification pathways (fm9, fc1). Farmers are more educated than ever but do not learn to have a system-oriented vision of agriculture (fc1). Furthermore, farmers-to-be follow education programs in specific schools, as from the age of 12 years old. They consequently develop a shared vision about farming mainly based on intensification, growth and high investments in equipment (fc1, fc2, fm3, fm7, fm9).

### 3.1.3. How Did the Interviewees Themselves Experience Lock-Ins in Their Own Trajectories and Pathways of Change?

We identified two pathways of changes. For some of the interviewees, quitting the mainstream dairy cooperative was a conscious decision to explore new ways of processing their milk (fm3, fm7, fm8, fm9, fm10). They were dissatisfied about the anonymity of contacts and the loss of control over the processing of milk in mainstream dairy cooperatives. For other interviewees, exploring an alternative pathway was a question of opportunity, either because a local cheese processor was looking for organic farmers (fm1, fm2, fm6) or because of the geographical proximity with a local cheese processor (fm4, fm5).

In five cases (fm3, fm4, fm5, fm6, fm9), changing trajectory also meant quitting a more intensive model in terms of production per cow. Others had already gone from an intensive towards a more extensive mode of production earlier on. They kept on adapting their farm to the requirements of cheese production within that trajectory (fm1, fm2, fm7, fm8, fm10). The interviewed farmers mention disapproval from other farmers (family members, neighbors, members of farmers' unions) when they decided to leave a mainstream dairy cooperative and process their milk in another way (fm7) or when they changed their way of farming towards more extensive practices (fm2, fm7, fm10). According to the interviewees, this shared vision orienting practices towards intensification is stronger in the "Pays de Herve", where less diversity regarding farm model exists in comparison with the "Haute Ardenne". One interviewee, from the Pays de Herve, chose to stop cheese processing on farm when she engaged in an intensification and growth pathway of her farm (fnc 2).

### 3.1.4. Did a Change in Trajectory Influence Their Approach of Farming Practices?

Many interviewees describe their change in pathway as satisfactory, because of a more stable remuneration (fm1, fm3, fm4, fm5, fm7, fm8, fm9, fm10) and a closer connection with the products processed with their milk. They also appreciate the human side of the connection with the local cheese processor (fm2, fm3, fm4, fm6, fm7, fm8, fm10). One interviewee (fm9) linked his differentiated vision about farming practices—no longer based on intensification and growing in scale—to the fact that he got the opportunity to deliver his milk to a local cheese processor. This example suggests that cultural conceptions are rooted in the organizational, technical and financial context in which farmers evolve.

Nevertheless, among the farmers, we also noticed that the idea of being a milk producer remained strongly rooted: the idea that they do not have the time or the competences to be involved in the processing of the milk was often expressed (fm2, fm3, fm4, fm6, fm9, fm10).

### 3.1.5. Interviewees Identified in Their History What Helped Them to Overcome the Obstacles

Interviewees cite three main factors explaining the success of their alternative trajectories, despite the lock-ins (Table 4):

1.　Family and network connections act positively on a change in path. Prior contacts with local cheese processors, for example through organic unions, are sources of opportunities for farmers (fm7, fc2). The implication of family members is an asset to process cheese on the farm or to invest time and energy in cooperative schemes with local cheese processors (fm7).

2.　Competencies and mentality are essential factors to succeed in alternative pathways. Interviewees recommend thinking out of the box and not listening to advice from others (fc1, fm9). The experience gathered outside of the agricultural world is an asset in terms of mentality

and acquired competencies (fm7, fc1, fc2). For this reason, one interviewee decided not to put his children in an agricultural school (fm7).

3.　　A positive feedback linked to the satisfaction reinforces the confidence in the trajectory of change.

**Table 4.** Factors identified by the interviewed farmers of the specialized dairy territories of the Walloon Region that helped them overcome the lock-ins preventing farmers to consider changes in pathways from the delivery of milk to mainstream dairy cooperatives, to cheese processing on farm or to the direct delivery of milk to a local cheese processor.

| Factors That Helped the Interviewed Farmers Consider a Change in Trajectory |
| --- |
| Social networks and the involvement of the family are sources of support and new opportunities |
| Ability to think out of the box |
| Experience gathered outside of the agricultural world |
| A positive feedback reinforces the confidence in the trajectory of change |

*3.2. Local Cheese Processors Also Experience Lock-Ins Acting against the Exploration of Alternative Pathways of Food Production*

3.2.1. Local Cheese Processors Experience Constraints Acting against Direct Milk Collection

Local cheese processors favor direct milk collection to control its features—taste, protein content, hygiene (cp1, cp2, cp3, fc1, fc2). Additionally, processing milk in a shorter timespan since milking guarantees a more stable protein configuration and increases the efficiency of milk processing (cp2). However, milk collection is costly and local processors do not necessarily find the ideal farmer nearby (cp1, fc1, fm3, fm4, fm6, fm7, fm8).

The milk collection policies of mainstream dairy cooperatives create a lock-in effect of an organizational nature against direct milk collection by local cheese processors. Mainstream dairy cooperatives do not tolerate variations in the quantity of milk delivered by a farmer (fnc1, fm2, fm5, fm10). Furthermore, mainstream dairy cooperatives do not see favorably that local processors collect milk directly from farmers. As local processors pay the milk at a better price, this raises the question of milk price paid to other farmers by mainstream dairy cooperatives (fm5) (When the milk price is high on the world market, some of the interviewed farmers note no substantial difference between the price they receive and the price given to farmers in the mainstream cooperative dairies (fm1, fm3, fm3, fm7, fm10). However, the price they receive remains stable, whereas the milk price drops in mainstream cooperatives dairies when the milk prices drop on the world markets (fm1, fm3, fm3, fm7, fm10). Some of the interviewed farmers (fm2, fm5, fm6, fm8, fm9) mention a price difference with the payments in mainstream dairy cooperatives that can amount to 10–15 cents/liter milk (fm8, fm9). Some local cheese processors pay better than others do (fm8, cp1). The possibility to discuss with the cheese processor and the balance of scale between the farmer and the cheese processor play a role in the milk price negotiation (fm5, fm8)).

The milk collection policies of mainstream dairy cooperatives leave the local cheese processor with two options. The first possibility is to collect the total production of one or more farmers. This can be a problem for small-scale local cheese processors, as they cannot ensure managing such a quantity of milk (fm10, cp3). The second possibility is to let mainstream dairy cooperatives supply them with milk. This option means relying on standardized milk for cheese production and losing control on the specific features of the milk. Local cheese processors overcome this lock-in by concluding contracts with newly created cooperatives of dairy farmers valorizing their milk on the European markets (cp3). The difference in size may affect the power of negotiation regarding milk price. It is also tempting for these cooperatives to conclude exclusive delivery agreements to bigger processors to the detriment of smaller ones.

### 3.2.2. Interviewees Consider That Their Small-Scale Businesses Face Distribution Pathways Not Adapted to Their Needs

Cheese production generates whey and cream (when the cheese processor uses skimmed milk (cp1, cp2, cp3)). The elimination of whey and cream is costly, and there is no market available for the small quantities produced (cp2). Calves and/or pigs can consume whey, and this is how farmers carrying out cheese processing on farm valorize this by-product (cp1, fc1).

The direct sale of cheese is not an option in most geographical areas covered: the location of farms or cheese-processing factories (fc2, fnc3) is remote and local consumers favor mass retail (fm9, fnc3, fm7). One farmer situated near an urban center developed direct sale successfully (fnc1). Some experienced sale on markets, which is very demanding in time and energy (fc2, fnc3). Price is an issue, as consumers remain mainly price-driven (fm8).

Local cheese processors mainly cooperate with generic wholesalers for the distribution of their products to specialized and mass retail. There is one wholesaler dedicated to small-scale organic productions. This wholesaler distributes products to specialized retailers and catering services. The interviewees feel uncomfortable in front of generic wholesalers focusing on quantities, promotional plans and price-driven competitiveness (fnc1, fnc2, fc1, fc2, fm7, fm8, cp3). Wholesalers are reluctant to collect small amounts of products, especially when the local cheese processors are geographically remote (fc1, cp1). The commercial relations with generic wholesalers are difficult (fnc2, fc1, fc2, cp3): there is an imbalance in power of negotiation (fnc2, fc1, cp3) and pressure on quantities and price (fc1, fc2, fm7).

When they upscale and produce larger quantities of cheese, cheese producers face requirements of mass retailers (packaging and promotional schemes) not sustainable for small-scale structures (fm7). Durable life date systems imposed by mass retailers are not always adapted to products like cheese, as cheese products gain gustative value by aging rather than worsening (fm7). When they upscale, local cheese processors rely more than before on generic wholesalers and mass retail. Some interviewees, therefore, prefer to remain small scaled and rely more on specialized distribution pathways (fc1, fc2).

### 3.2.3. Interviewees Identify the Elements that Might Alleviate the Constraints on Their Businesses

The interviewees cite two main factors contributing to the success of their trajectories of cheese processing:

1. Experience in business matters outside of the agricultural world provides competencies in management (fc1).
2. Interviewees appreciate the existence of a dedicated wholesaler specialized in organic, small-scale farm productions. This wholesaler makes access to specialized retailers easier and less time-consuming (fc2, fnc1). Interviewees appreciate not having to lose time and energy on marketing issues (cp1). They would like specialized retailers to emphasize more on local cheese production (cp3, fm7).

Interviewees consider that more organization among local cheese processors would be useful to defend their interests (cp1, fc1, fc2, cp3). By the time of the study, there was no collective organization to promote small-scale non-industrial cheese productions. Interviewees mention a general mentality not oriented towards collective action in the concerned territories, in opposition to other European countries where farmers and local processors were more collectively organized (cp1, fc1, fc2, cp3).

## 4. Discussion

### 4.1. Our Study Identifies a Set of Coherent Lock-Ins Limiting Alternatives Pathways of Farming and Milk Processing

We identify in our results a relation of reciprocity between the farmer, their practices and visions about their practices, and the local cheese processor, or the cheese-processing activity on farm.

Local cheese processors wishing to collect milk directly are dependent on the existence of farmers capable to meet their requirements. On the other hand, farmers will not be encouraged to maintain a farm model meeting the requirements of cheese processing if no perspective in this direction is present.

Besides technological, cultural and 'knowledge-driven' lock-ins, this study brought forward a type of lock-in we call 'organizational'. The way actors organize/structure themselves in the broader dairy context (mainstream dairy cooperatives, educational and counseling systems, public policies, banks, retail and distribution, consumers—we define these actors and the way they organize themselves as "mainstream dairy context") leads to the disqualification of other ways of farming and of processing food.

The results draw the picture of a mainstream dairy context structured with coherence. This coherence limits the potential of differentiated ways of creating and processing milk (Figure 2).

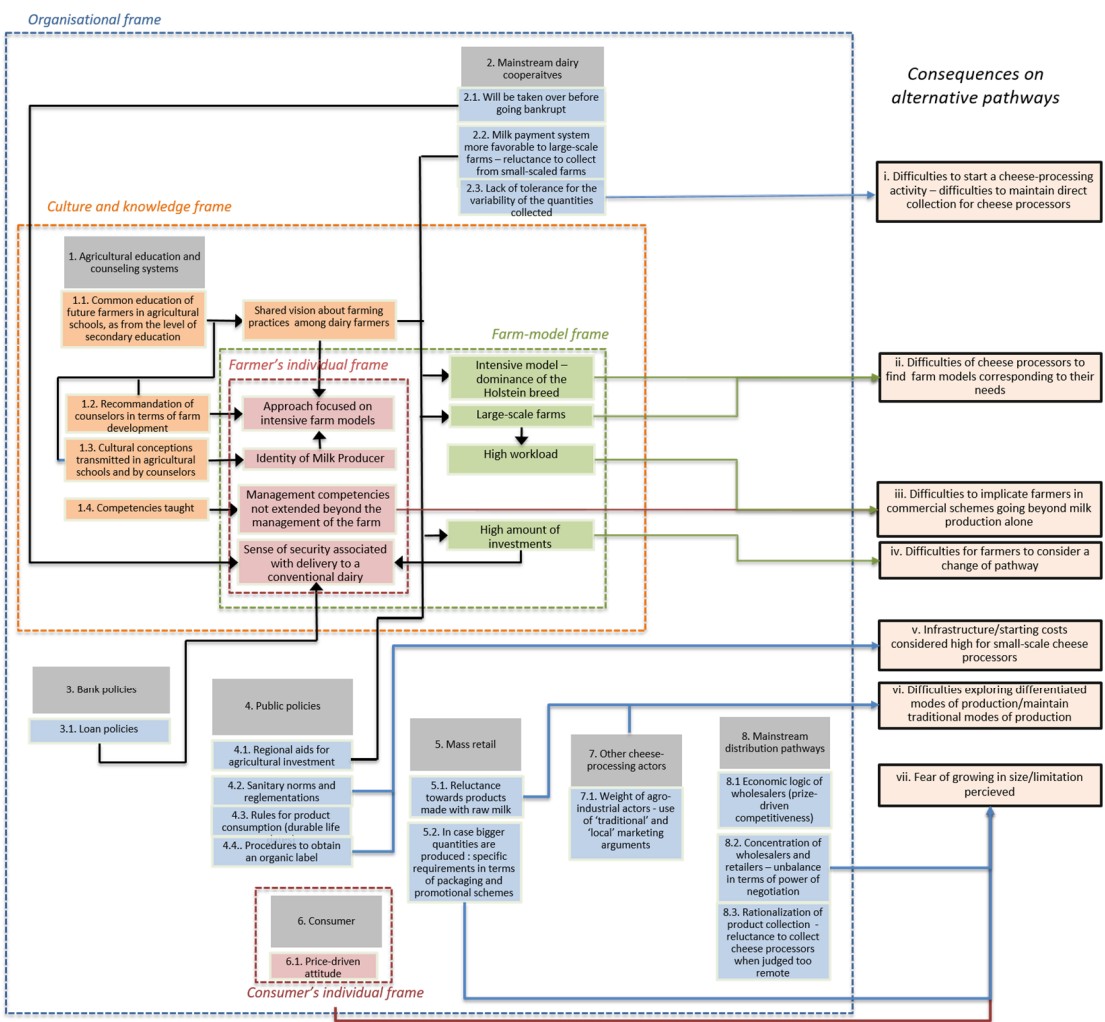

**Figure 2.** The coherent action of lock-ins on cheese processing alternatives.

At the farm level, the needs of the mainstream dairy cooperatives for standardized milk and the subsequent payment strategies orient the farm model. The organization of the distribution pathways and the consumer's attitude make local cheese processors limit their market approach. De Greef and Casabianca [24] describe a similar sectoral structure in the Dutch pork chain, driven by commodity-logics and standardized quality. Diversification towards less "standard" productions fails "because of price effects" and a reluctance of processors and of the retail sector to consider and support alternatives. They similarly notice a direct consequence of this organization on farms, lead on "an industry-driven route of increasing size and efficiency". De Greef and Casabianca [24] in the case

of the Dutch pork chain and Fares, Magrini and Triboulet [28] in the case of the French wheat supply chain stress the non-integration of the value chain, that is the absence of a link between farmers and the downward processing structures. These authors attribute to the non-integration of the value chain the difficulty to consider and support strategies for change. Concerning our case study, we might notice that the milk sector seems more integrated than the Dutch pork chain [24] or the French wheat supply chain [28]. Mainstream dairy cooperatives hold a vertical link between dairy farmers and the milk processing structures. Nevertheless, their present configuration leaves the farmers with little strategic power [8,80].

The coherence of the mainstream dairy context seems to be a good illustration of a locked-in socio-technical regime [18]. This socio-technical regime seems to have followed a path of co-evolution: public policies, educational systems and consumers' behavior are in line with the agro-industrial pathways of milk processing and distribution. The organization of the socio-technical regime orients the farm model and constrains the ability of individuals to act on an alternative paths. In the case of the French wheat supply chain, Fares, Magrini and Triboulet [28] described a supply chain strongly concentrated downward at the farm level. They stressed that this concentration generated structural lock-ins: downward concentrated actors have a power of negotiation over other actors and use inter-professional agreements to impose production standards. Upward actors, especially farmers, have little space left to engage in alternative production or transformation pathways, and if they do so, have to support significant personal risk. Our study reveals similar lock-ins concerning the Walloon mainstream dairy context. Local cheese processors and farmers delivering milk directly to them evolve in a relation of reciprocity. They experience lock-in effects, tending to make them move away from that reciprocity. There is a reinforcing effect of the mainstream dairy context against alternative ways of processing milk.

The impact of this context is not constant over the studied territories. Small-scale extensive farms still present in the territory Haute Ardenne may more easily answer the requirements of local cheese processors. This resonates with what Morgan et al. [51] and Murdoch et al. [51] noticed: not all environments present the same "ecological conditions" for the development of alternative models of food production. Territories "that have not been fully incorporated into the industrial model of production" [81] or "where opportunity for large-scale, intensive and industrial farming has been restricted" [51] are more likely to host a greater diversity of farm models and, hence, to host differentiated food systems.

*4.2. The Locks-Ins Embed the Farmer's Frame in the Organizational Frame of the Mainstream Dairy Context*

If we consider the agency of farmers, this case study reveals how a set of lock-ins belonging to the farm-model frame and the more general cultural and knowledge frame determines the farmer's individual frame, regarding competences, identity or the consideration of risk. The organizational frame of the mainstream dairy context embeds both frames (Figure 2).

Figure 2 stresses the embedded aspects of path dependency: the interactions between frames equip actors with competencies in line with the needs of the socio-technical regime. At the farmer level (Figure 3), organizational lock-ins contribute to orient farmers towards large-scale farm models, whose practices contribute to feed the identity of the farmer as milk producer. The farmer defines themselves as such and reinforces in turn their potential of action within the coherence of the mainstream dairy context. Our results illustrate that path dependency involves a process of interaction between collective and individual frames: agents are embedded into the coherence of the socio-technical regime and contribute, through their actions, to the further coherence of the regime in which they evolve [36].

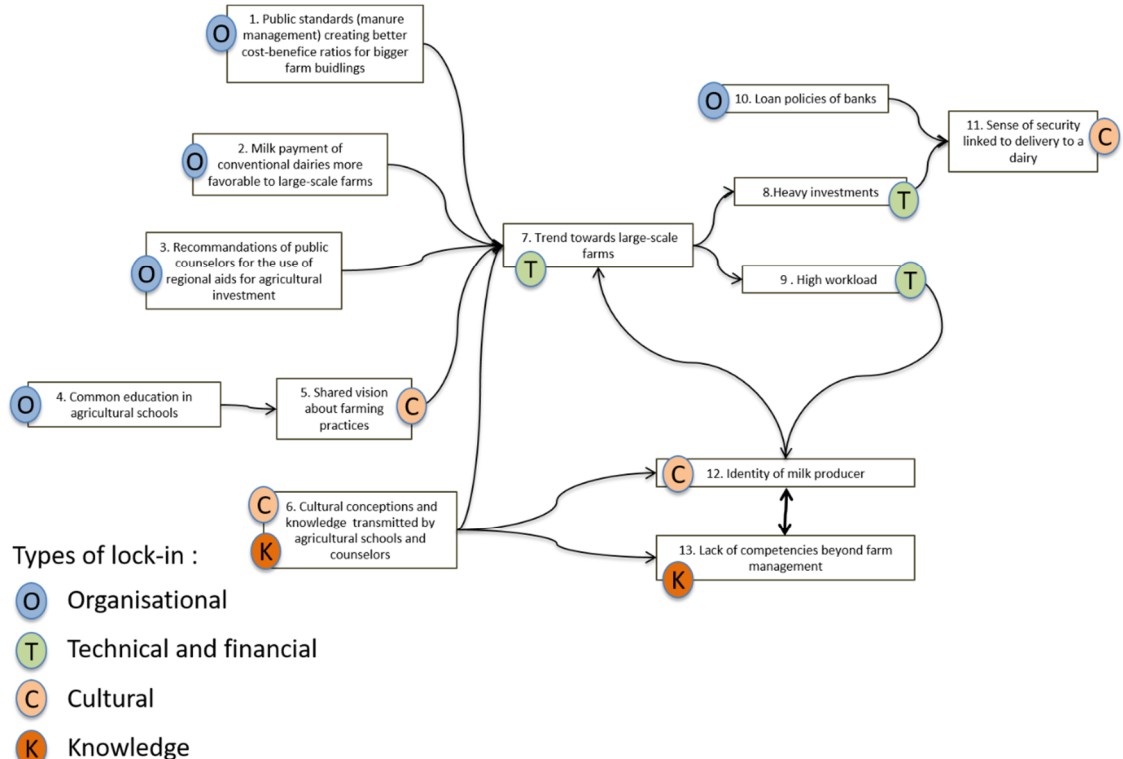

**Figure 3.** Lock-ins acting and reinforcing themselves at the farmer level.

### 4.3. Our Study Identifies Agency at the Crossover of Top–Down and Bottom–Up Processes

Our study stresses how the wider organizational frame of the mainstream dairy context embeds the farm model and the farmer's individual frame (Figure 2). One interviewee (Results, Section 3.1.4) draws a link between the evolution of his vision about what he sees as "good farming practices" and his experience of milk delivery to a local cheese processor. Such a phenomenon, also described in other case studies [44] and theoretically discussed [49,50], suggests that change in farming practices can lead an individual to perceive differently the farming context in which he evolves and question the cultural lock-in he had previously integrated.

Our findings suggest that we might foster changes in farming practices, and hence in the farmer's approach regarding farming, by supporting agro-food supply chains based on a differentiated milk quality. Support to differentiated food chains has to take into account the need for dedicated services in terms of distribution of products, risk management and adequate representation. The interviewees note that the mainstream distribution systems are not adapted to their needs and hold features of an imbalance of power due to the concentration of actors present (results, Section 3.2.2). They call for the development of a network of wholesalers and retailers more dedicated to local and small-scale production. If indeed the market turns out to be an "obligatory passage point" as stated by Renting and Marsden [82] citing Callon [83], it will be necessary to organize this "passage point". Beyond collective representation (see results, Section 3.2.2) this probably calls for a reflection on the appropriate networks to develop, going beyond the sole—often studied [72]—direct distribution networks [84,85].

In terms of public policy, our study stresses that alternatives rely on specific farm models. Defining support policies guaranteeing the persistence of a diversity of farm practices may, on a long-term basis, prove beneficial as support to a greater variety of types of rural development.

Finally, at the individual level, which factors allowed interviewees to exercise agency in favor of change despite the existence of lock-ins? We identify five factors:

1. The ability to question the shared vision about farming practices among dairy farmers;
2. The ability to stand against reprobation from neighbors and family members;

3.   Competences going beyond farm management solely;
4.   A familial implication in the farming-related business;
5.   The resort to a prior network of connections.

Previous case studies also identified these factors as drivers for change [54] (factors 1, 2, 3, 5) [55] (factor 4). In more theoretical articles, authors also stressed the importance of knowledge as a source of individual power [86] and the interpersonal network around the individual as a source of adaptability and resilience [56].

When we consider the education of farmers, as described by the interviewees (see results, Section 3.1.2.), we understand that its purpose is to equip farmers with a strong technical background. This logic makes sense in the view of the national and European agricultural policies as they have been defined throughout the twentieth century [56]: gathering farmers together from a young age can ensure the integration of common standards and practices. Our results suggest that a modification of educational policy might be favorable to a greater adaptability of farmers today:

1.   In terms of content: adaptability depends on management competencies beyond the technical aspects of farm production or farm management. Would it not be relevant to integrate these elements in the educational programs? Do programs sufficiently equip dairy farmers in terms of capability and adaptability?
2.   In terms of organization: would an education of farmers less separated from other professions not allow greater openness to competences and networks that might prove useful concerning their adaptability to a changing environment?

This study invites us to consider the role of agency in transition processes as a dialectic process at the crossover of the individual's or network's capabilities and structural changes in the organizational and cultural environment. In this regard, our study ties up with the most recent theoretical discussions on how to approach processes of change [39,87]. Generally [87], and in the agro-food sector [39], change is a constant co-evolution of top–down and bottom–up [87], and "diffuse and intermingling" [39] processes.

The identification of the link between agency and structural dynamics emerged from an assumed methodology putting the emphasis on the study of individual trajectories. The study revealed a web of context-linked features whose significance goes beyond the contingencies of individual trajectories. Indeed, the trajectories taken as a phenomenological lens [75] not only disclosed characteristics of the mainstream dairy context in line with previous studies on the agro-food sector [24,28,44,51,54–56,81], but they also revealed the grip of the context on individual trajectories. The combined comprehension of the web of convergent and interconnected lock-ins and of the way actors managed to overcome lock-ins holds a significance that goes beyond the particular trajectories of actors. This research calls for a further and broader inquiry on the contextual embeddedness of the identity and strategic choices of farmers.

## 5. Conclusions

The analysis of alternative pathways of milk processing revealed convergent and interconnected lock-ins originating from the mainstream dairy context. Our study stresses the strength of lock-ins on the agency of actors. The interconnectedness of lock-in goes from the organizational frame of the socio-technical regime to the capabilities and identities of actors. Our study stresses that the organizational frame of the agro-food regime influences farm practices and that local processors may support another evolution of farming models. Pathways of transition might be favored by acting on the organizational lock-in present, at the level of the education of farmers and in the organization of the distribution pathways.

Our approach mobilizes a combination of evolutionary approaches on transition and considerations on individual pathways of change. The Multi-Level Perspective states that alternatives develop through the emergence of protective spaces called niches [17]. Rather than a niche configuration, our study revealed the embeddedness of alternatives into the environment in which they emerged.

The embeddedness affects how individuals perceive their environment and has consequences on the opportunities that actors may seize and on which personal resources they may mobilize. Rather than endorsing a deterministic approach about agency, our study stresses that individual empowerment is a matter of connections, experience, and education, and that drivers for transition lie at the crossover of actors' empowerment and systemic change.

**Author Contributions:** Conceptualization, V.D.H. and P.V.B.; Methodology: V.D.H. and P.V.B.; Formal analysis, investigation and resources: V.D.H.; Writing: V.D.H., K.M. and P.V.B.; Supervision: K.M. and P.V.B.

**Funding:** This research was conducted with the financial support of the Université catholique de Louvain and of the Fonds de la Recherche Scientifique—FNRS—Fonds pour la Formation à la Recherche dans l'Industrie et dans l'Agriculture-FRIA.

**Acknowledgments:** The authors are grateful to Thérésa Lebacq (Université Libre de Bruxelles) for co-supervising the project design and Kevin Morel (Earth and Life Institute—Agronomy, Université catholique de Louvain, Belgium) for helpful comments on earlier versions of this paper. However, the analysis and comments remain our responsibility solely.

**Conflicts of Interest:** The authors declare no conflict of interest.

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
