# Peer review of "Lock-ins and Agency: Towards an Embedded Approach of Individual Pathways in the Walloon Dairy Sector"

_sustainability, doi:10.3390/su11164405_

Round 1

Reviewer 1 Report

The manuscript is not suitable for publishing. It is not well written and hard to read. It is not clear for its research objectives and findings.  I can’t see any academic contribution and innovation for sustainability from this study. The sample size is too sample, even for a qualitative study to get meaningful scientific results.

Reviewer 2 Report

This manuscript aimed to design a qualitative case study based on interviews with farmers representing 75% of the revenues of the Walloon region with the goal of understanding an objective that is unclear to me as the reviewer. The objective is either using an interview to understand alternatives available to farmers for processing milk as opposed to shipping to the cooperative (as mentioned in the abstract), Or “We propose a qualitative study based on semi-directed interviews with farmers and cheese processors active in cheese-processing alternative trajectories in the Walloon dairy territories”as stated in the introduction in line 101-102, as mentioned in the introduction. The authors concluded the farming model is evolving from grazing systems towards intensive farming, though individual identification is important to farmers, and that movement towards ag education will enable farmers to produce change favorable to their industry.

 While the manuscript presents a topic that has not yet been widely discussed, the manuscript cannot proceed until it is revised for the ease of reading, especially in not using consistent terms. For example, processors are mentioned, but then later referred to as dairies, dairies are referred to as cooperates and farmers are also dairies. It makes it very difficult to understand the introduction, and even the objective, which I still cannot grasp due to a lack of clarity, and the authors need to pick terms and stick with them throughout the manuscript. In addition, many adjectives are added to a sentence that provide no additional information, and I believe this will make any reader have difficulty following. Therefore, a major focus in this revision should be deleting unnecessary information that is irrelevant to the sentences. Citations are missing in several places, many terms are improperly written and a simple read through would have caught them (he in reference to a dairy, far in reference to farm) and the adjectives added make the manuscript vague and difficult to follow.  For example, the authors phrase their main argument for their study objective in this manner. “From this rather irrevocable diagnosis, raises the question of whether dairy farmers  might consider alternatives to the delivery of milk to dairies, and, if attractive alternatives there are,  why there aren't a large number of farmers tempted to opt for them.” 

This is the selling point of the manuscript and yet, it is lost in fluffy adjectives. I highly recommend streamlining this manuscript for ease of reading.

One of the largest concerns I have about this study is the scientific merit of this manuscript, the objective is very unclear, I still do not know which objective pertains to the study as they say many different things in different places.

It reads in the abstract, “Our objective was to understand the issues faced by these actors in relation to the  mainstream dairy context from which they emerge.” Since this was unclear, I thought the objective was based on a preluding statement to the objective in the abstract. “We decided to analyze the trajectories of Walloon dairy farmers exploring alternatives to the delivery of milk to these mainstream dairy cooperatives.” 

However, I reviewed the entire introduction and still do not understand if this is about farms considering alternatives for milk processing as opposed to selling milk to a cooperative, or if this interview was directed towards farms already participating in direct on farm cheese processing as the introduction eludes to “We propose a qualitative study based on semi-directed interviews with farmers and cheese processors active in cheese-processing alternative trajectories in the Walloon dairy territories.: In line 101-102.

I continued this review under the assumption that the objective is about current farmers using on farm cheese processing based on the questions asked of the farmer in the interview. However, In lines 188-190 the objective reads: “Our study intends to focus specifically on the strategies of farmers, who are exploring alternatives, not regarding their farm model, but regarding the milk payment and valorisation of their milk.”

Thus for scientific merit, clarity is needed for the objective, and the introduction needs to be streamlined to meet the true objective of the study as both potential objectives mentioned were covered in the introduction. This is a major weakness and makes the study difficult to review.  

Specific comments are as follows:

Objective: Authors, what is the objective of this study? To interview farmers already involved in cheese processing or those considering to switch and understanding why they haven’t? I have no idea what your target audience is since you state both objectives as your objective. Please pick one objective and consistently reference it throughout. I stopped reviewing after the introduction so that I could appropriately review your materials and methods based on the true objective.

L14-19 For the broader scope of the audience (ie USA, Australia etc.) and for understanding, please include a figure of  where the Walloon region is located, especially in relation to the territory of the milk collection for these 4 cooperative territories, and highlight the region where the areas where focused on for “east of the Walloon region representing 75% of the farming revenues.”

L17 “Our objective was to understand the issues faced by these actors in relation to the mainstream dairy context from which they emerge.” This sentence is extremely wordy and vague. Since your objective should be the most clear part of the manuscript please rephrase. Please also include mention of  “understanding the alternatives of delivery of milk to these cooperatives” in your objective as this was part of why you conducted the research.

L24-25 This is very wordy, please rewrite to be more simple and clear for the reader

L36-38 “Following the 2003 Luxembourg Agreement reforming the Common Agricultural Policy,

 the European Union made plans to abandon the milk quota system at the horizon 2015, gradually  reduce intervention prices and decouple subsidies from milk production.” citation is needed to support this statement

The paragraph “The price shock put the farmers in financial difficulties and exacerbated tensions between 43  farmers and their  dairies” is missing a key point. Farmers weren’t just frustrated with their dairies, they were frustrated with the milk price being paid by the processors. Perhaps when you use the term you mean dairy? I would use processor as done in the abstract. You must discuss this connection as price (as mentioned above) was why farmers dumped milk on the fields, it cost more to ship the milk than it did to receive the paycheck for it. Please include that connection, as of right now, that paragraph looks like farmers dumped milk because they were directly upset with their dairies.

L43-44 “The price shock put the farmers in financial difficulties and exacerbated tensions between 43  farmers and their  dairies” This heading should include mention of the tensions of farmers between processors as well no? Is it not the processor who pays the farm as well?

L45-46 “At the time of the dairy crisis of 2009, angry dairy farmers shedding milk on fields, streets and  public institutions received extensive media coverage.” Please use citation to support this statement 

L52 In the “specialized dairy territories of the Walloon region…” Why are they specialized dairy regions are they known for producing cheese or a specific product? Please explain how the territory is specialized

L54 Earlier you said intensification, now you are calling it concentration trends. Please pick one term to refer to larger herd size and stick to it, or if that is not what was intended here please rephrase and be more clear.

L54-56 “Greatly influenced by the concentration trends, the milk sector in the Walloon region is mainly

 organized around four dairy cooperatives, which are all members of the Belgian Confederation of dairy industries.” Needs to be cited 

L64 “(1900 cooperates over wallonia)” is unclear are these 1900 dairies, or something else? Clarify

L67 if 12000 cooperates refers to dairies please call them dairies for clarity or specify

L71-72 How did the farmers express their distress? Either use a quote to support your findings, or describe the distress in a more clear manner. 

L72-73 “From this rather irrevocable diagnosis, raises the question of whether dairy farmers might consider alternatives to the delivery of milk to dairies, and, if attractive alternatives there are,  why there aren't a large number of farmers tempted to opt for them.’ This is a major selling point for why you conducted your research and right now the vagueness of the sentence with lots of adjectives that aren’t necessary is losing the message. For example,  “from this rather irrevocable diagnosis, raises the question” is completely unnecessary. Please make this statement very clear for the reader, so the message is received that you are building the case for your research here. 

L82-83  “Following the intensification of farms and the strategic choices of milk valorisation taken by dairies, cheese processing concentrated by one cheese processor.” Rephrase this doesn’t make any sense, and part of that is because terms are made interchangeable between processing and dairies and processor. It is so unclear.

L84-85 “He successfully introduced a protected designation of origin for what concerns one of the three territories, the PDO "Fromage de Herve." Who does he refer to, be specific? Is this a person? Spell out PDO as this is not a generic term accepted by this journal. Please also clarify this is about cheese. 

L86 Please refrain from assigning gender to the processor in the manuscript

Reviewer 3 Report

The article is conceptually very interesting and poses a couple of very relevant socio-economic questions for the dairy supply chain as a whole. It is quite thorough and well-presented, exposing several problems from various perspectives. 

I have a couple of remarks that I think would help to improve the transmission of your message. 

Introduction

The Introduction as a whole is too long, the idea is to present the current research and your research objective. In my opinion, there is too much conceptual framing of the current research and comparing it to yours (this could perhaps partially be done in the Discussion). The description of the 2009 milk crisis and the Wallon dairy sector is also too extensive and descriptive, I propose to condense it to a shorter, more streamlined version. I also suggest deleting the subtitles in the Introduction because they interrupt the flow of the text. 

Materials and methods

Could you provide more information on the total size of the sample? A table here that shows the information would be nice. 

Results

Table 2. needs to be sourced and mentioned in the text. Also, no factors that help farmers in the change of trajectory are revealed for “Public agricultural advisers and banks support farming practices based on intensification, growth and high investment”.

Discussion

Figure 1-You wrote prize-driven attitude instead of price-driven for the consumer. Just to differentiate better, I think it would be good to write conventional or pre-established pathway for the conventional dairies.

Figure 2 seems quite repetitive. Could it be embedded in Figure 1? 

I am also wondering if you could discuss a little bit about the opportunities for the farmers in establishing alternative pathways for their milk? What are the potential benefits (e.g. more freedom, independence, and flexibility) and ways in which they could establish more alternative pathways in the future? It would be good to highlight the potential that unconventional pathways can bring for the future. Being more proactive in raising the awareness of the consumers could also play a role here. Perhaps creating new distribution channels where consumers buy directly from the farmers and the cheese producers, increasing the brand value and customer loyalty, and getting out of the solely price-driven behavior. If there are some studies on this topic, it would be interesting to discuss them briefly. 

General remarks 

Language is overly wordy making it difficult to read at times. Some of the language constructions are incorrect (e.g. line 72-74). I propose to have the whole article proofread for better readability after making all the adjustments. 

I am not sure if the word valorization is properly used here; do you mean pricing or imposing restraints and fixing the price (valorization)? Or you actually mean valuing? 

Round 2

Reviewer 1 Report

Where is your point-to-point response letter to address each line of my comments?

Reviewer 2 Report

This manuscript has made remarkable improvements in clarity, as well as readability. Thank you to the authors for addressing my comments in regards to the clear objectives, as well as adding clarity to the introduction, and adding more details about which farmers were interviewed, as well as providing a more clear version of this manuscript for the reader. It is quite clear a lot of time was placed into improving the manuscript as the materials and methods are vastly improved, and now that the writing is in a better place, the manuscript needs minor adjustments.  Since this interview was geared towards Belgian farmers, I would suggest to the authors to rephrase the title from “dairy sector” to include Walloon region, or even just Belgian dairy sector since this interview was not conducted on all dairies, just those specific to this region. The authors still need to also reformat the tables 

Tables: All of the tables have different formatting. Please review guides for formatting tables for this journal and fix accordingly. 

All table descriptions are too vague. Please make the labeling in the tables specific so that they can stand alone. For example, “Table 1.  Qualitative sample distribution for an interview investigating challenges for farmers in the Walloon Region who process milk on farm, deliver milk directly to a local cheese processor, or local cheese processors which collect milk directly from farms”

The manuscript will benefit greatly from a heading such as this that is replicated for each table so they stand alone. 

Table 1 has spacing in the headings which needs to be fixed so that it easily reads.

Heading “Local processor” should say “local cheese processor who collects milk directly from farmer”

Table 2 Is centered as opposed to left aligned. Please adjust so that the table is left aligned. Please also be consistent with lining of the tables. Some lines in this table go across and others do not. Make changes so the Table is similar to Table 1. 

Table 3 Also centered, please use left alignment. Please review guidelines for which centering is appropriate for this journal and make all tables consistent with formatting, including legible headings and consistent line spacing. 

Table 4 Wrong alignment 

Specific comments:

L14 Specify Walloon Region of Belgium (for the lay reader that is not part of the EU). 

L148 The introduction is much approved and does a great job of laying out why farmers were interviewed, and also chosen for the lock-in sector approach. However, the objective written in your abstract (L19-22) needs to be restated at the end of the introduction. Please add so that it is specific and clear to the reader at the end of the introduction. 

L163 Please fix grammar

L223 Specify what the local cheese processors requirements for farmers are (gustative quality of milk, protein, hygiene)

L249 Identified

L299 “Additionally, processing milk in a shorter timespan since milking guarantees an optimal protein configuration”

This sentence still reads unclear, please re-word. Do you mean protein configuration changes based on frequency that cows are milked, or do you mean that shorter pick up times results in more stable protein profiles of the milk?

L303 Message of this sentence will be more clear if you remove “less than before” from this sentence

L307 Where you able to determine how much more per L a farmer was paid for the local scale cheese processing as opposed to the traditional cooperatives? If not, you cannot make this statement. While it is very likely true, you can only make that statement if you have the evidence to support it. If this is true, then you need to state the average price differential paid by local processor versus the cooperative at that given time that the interviews were conducted.

L319-321 Serum is not the correct word choice here. Do you mean whey as a by-product of cheese processing? Similarly, cream is not typically considered a by-product either, especially to local cheese processors as it is considered to be of high value. 

Please revise

L424 Please include a citation here

L465 Specify which of the 5 factors were drivers for change in the citations mentioned for farm diversification in citation 54 and Danish farmers in citation 55

L466 You mention citation 54, and 55 as “they” citing knowledge as a “source of individual power,” and “source of adaptability and resilience, but do not cite 54 or 55 here. Rather than stating “they” be more specific  to which reference you are referring to , and also please re-cite them in the sentence to be clear. Right now it is very unclear.

L484 What do these theoretical discussions suggest? Do they suggest we create alternative pathways for farmers, despite lock-ins, or do they suggest we adjust education for the agricultural community as a whole to create more flexibility in their thinking? Please specify what the citations suggest

L490 Please include citations that go along with studies in the ago food sector. 

L506 citation needed here. Also Niche doesn’t need to be capitalized

Round 3

Reviewer 1 Report

Thanks for your detailed responses well addressing my comments.